# Balance of Macrophage Activation by a Complex Coacervate-Based Adhesive Drug Carrier Facilitates Diabetic Wound Healing

**DOI:** 10.3390/antiox11122351

**Published:** 2022-11-28

**Authors:** Ching-Shuen Wang, Shen-Dean Luo, Shihai Jia, Wilfred Wu, Shwu-Fen Chang, Sheng-Wei Feng, Chieh-Hsiang Yang, Jiann-Her Lin, Yinshen Wee

**Affiliations:** 1School of Dentistry, College of Oral Medicine, Taipei Medical University, Taipei 110, Taiwan; 2Department of Otolaryngology, Kaohsiung Chang Gung Memorial Hospital and Chang Gung University, College of Medicine, Kaohsiung 833, Taiwan; 3Department of Neurobiology, University of Utah, Salt Lake City, UT 84112, USA; 4Department of Genetics and Genome Sciences, School of Medicine, Case Western University, Cleveland, OH 44106, USA; 5Graduate Institute of Medical Sciences, College of Medicine, Taipei Medical University, Taipei 110, Taiwan; 6Department of Oncological Sciences, University of Utah, Salt Lake City, UT 84112, USA; 7Department of Neurosurgery, Taipei Medical University Hospital, Taipei 110, Taiwan; 8Department of Pathology, University of Utah, Salt Lake City, UT 84112, USA

**Keywords:** complex coacervates, oligochitosan, phytic acid, antioxidants, reactive oxygen species, sustained release, diabetic wound healing, M2 macrophage polarization, db/db diabetic model

## Abstract

Uncontrolled and sustained inflammation disrupts the wound-healing process and produces excessive reactive oxygen species, resulting in chronic or impaired wound closure. Natural antioxidants such as plant-based extracts and natural polysaccharides have a long history in wound care. However, they are hard to apply to wound beds due to high levels of exudate or anatomical sites to which securing a dressing is difficult. Therefore, we developed a complex coacervate-based drug carrier with underwater adhesive properties that circumvents these challenges by enabling wet adhesion and controlling inflammatory responses. This resulted in significantly accelerated wound healing through balancing the pro- and anti-inflammatory responses in macrophages. In brief, we designed a complex coacervate-based drug carrier (ADC) comprising oligochitosan and inositol hexaphosphate to entrap and release antioxidant proanthocyanins (PA) in a sustained way. The results from in vitro experiments demonstrated that ADC is able to reduce LPS-stimulated pro-inflammatory responses in macrophages. The ability of ADC to reduce LPS-stimulated pro-inflammatory responses in macrophages is even more promising when ADC is encapsulated with PA (ADC-PA). Our results indicate that ADC-PA is able to polarize macrophages into an M2 tissue-healing phenotype via up-regulation of anti-inflammatory and resolution of inflammatory responses. Treatment with ADC-PA around the wound beds fine-tunes the balance between the numbers of inducible nitric oxide synthase-positive (iNOS+) and mannose receptor-negative (CD206-) M1 and iNOS-CD206+ M2 macrophages in the wound microenvironment compared to controls. Achieving such a balance between the numbers of iNOS+CD206- M1 and iNOS-CD206+ M2 macrophages in the wound microenvironment has led to significantly improved wound closure in mouse models of diabetes, which exhibit severe impairments in wound healing. Together, our results demonstrate for the first time the use of a complex coacervate-based drug delivery system to promote timely resolution of the inflammatory responses for diabetic wound healing by fine-tuning the functions of macrophages.

## 1. Introduction

Diabetes is an increasing healthcare problem worldwide. It is estimated that there are over 6.5 million patients impacted by this disease, incurring yearly costs that exceed $245 billion [1]. One of the most common complications of diabetes is chronic wounds [2]. Such non-traumatic wounds precede 85% of all diabetic lower extremity amputations, and once amputation occurs, patients have a five-year mortality rate of 50% due to diabetes-related postoperative wound complications [3]. It has been suggested that chronic wounds in diabetics can be caused by uncontrolled and sustained inflammatory states with elevated levels of pro-inflammatory cytokines and reactive oxygen species (ROS), which in turn impair the expression of growth factors for normal healing [4,5,6]. Therefore, developing new strategies to control inflammation that reduces the overall level of oxidative stress in diabetic wounds is a high priority.

Wound healing and repair involve a complex sequence of cellular and molecular processes, including inflammation, cell proliferation, angiogenesis, collagen deposition, and re-epithelialization [7,8,9]. One of the earliest events of a wound-healing response is the infiltration of inflammatory cells at the wound site [9]. The infiltration of inflammatory cells following tissue injury is a critical contributor to wound healing. However, prolonged inflammatory responses involve the formation of excessive ROS, with failed resolution in wounds known to favor the polarization of M1 pro-inflammatory macrophages while impairing the polarization of M2 macrophages that promote the wound-healing process [10,11,12,13]. Increased M1:M2 macrophage ratios are found in diabetic wounds, resulting in a failure to progress in the cellular and molecular processes of wound healing, which eventually leads to chronic wounds [11]. Thus, efforts attempting to apply biomaterial-based drug delivery systems to treat macrophage dysfunction in chronic diabetic wounds are increasing. Specifically, when damaged tissue interacts with biomaterial-based drug carriers, it is desirable to modulate the phenotypic changes of macrophages from pro-inflammatory M1 to M2 tissue healing [14].

M1 macrophages dominate wound beds in diabetic patients over M2 macrophages [11,12,14], while dressings that can hold drugs in place in wet environments and reduce the time between dressing changes also remain a great challenge faced by chronic wound treatment in diabetic patients. Therefore, we have developed a new type of diabetic wound dressing material based on complex coacervation, which could provide a sustained drug release effect within wet environments and modulate macrophages’ behavior, thus accelerating diabetic wound closure. Complex coacervation plays a role in the formation of the underwater adhesive of the Sandcastle worm (*Phragmatopoma californica*) [15,16,17]. The formation of complex coacervation is a phenomenon of liquid–liquid phase separation, whereas the dense phase contains oppositely charged molecules and is separated from the surrounding diluted phase. Such a phenomenon is broadly applicable to endowing biomaterials with underwater adhesive properties, which is immiscible within a fluid environment and therefore particularly suitable for wound dressing [15,18]. Using a complex coacervate-based drug delivery system is a relatively new concept [19,20,21]. Although its use in medicine is more recent, it has been widely applied in the food industry and for personal care since the early 20th century [22,23,24]. The advantages of using a complex coacervate as a controlled release system are significant [21,25]. For example, it provides in situ formation in the aqueous environment without a toxic solvent and shear thinning property, making it suitable for injection. Moreover, it can also be self-assembled into a highly dense liquid phase, which can provide protection of labile bioactives, as well as a controlled release of drugs in the targeted tissue with fewer side effects [20].

Currently, there are only a handful of studies using a complex coacervate to engage the host function for tissue healing [26,27,28]. For example, heparin-based coacervate complexed with growth factors (e.g., fibroblast growth factor 2, epidermal growth factor, and bone morphogenetic protein-2) has been shown to promote tissue repair in several disease models, such as infarct myocardium, normal wounds, and bone repair [26,27,28]. Other examples have used positively charged materials to deliver plasmid DNA, microRNA, and siRNA for gene delivery in treating atherosclerosis and cancer [20]. Although these studies provide invaluable new insights into using a complex coacervate in the field of drug delivery and tissue regeneration, the underlying molecular mechanism of the interaction between innate immunity and implanted coacervate-based biomaterials remains largely unexplored.

Natural antioxidants have a long history in wound care [29]. For example, it is well-known that PAs significantly reduce ROS levels in various models [30,31,32]. It also shown that PAs can inhibit the production of an array of cytokines by suppressing the NF-kB signaling pathway [33,34,35]. Such antioxidative stress and inflammatory properties of PAs are important during wound healing. However, they lack clinical feasibility due to their complicated application to wound beds due to high levels of exudate or anatomical sites to which securing a dressing is difficult. This study aimed to retain the safety and benefits offered by using natural antioxidants for wound healing while addressing clinical feasibility when designing a delivery system. In this study, two natural compounds were used to generate a complex coacervate-based drug carrier (ADC) due to their potential health benefits observed in both in vitro and in vivo studies. One of them is the highly positively charged oligochitosan (Och), which has been widely used in the field of drug delivery due to its intrinsic high water solubility, biocompatibility, and biodegradability [36,37,38]. The counterpart of Och is highly negatively charged antioxidant inositol hexaphosphate (IP6, known as phytic acid), which has attracted significant attention due to its potential in anti-inflammatory responses, ROS-inhibition, and capability to polarize macrophages into an M2a resolution phenotype [39,40,41]. Although Och and IP6 have been widely studied, this is the first study that aims to combine these promising natural compounds to create a new type of drug carrier based on complex coacervation with underwater adhesive properties and investigate its interaction with macrophages. Here, we present evidence that ADC could be used as a unique drug carrier to encapsulate and protect antioxidant proanthocyanins as a sustained release platform to improve skin wound healing in diabetic mice. The overall result of diabetic wound treatment with this one-time and easy-to-make bioactive compound delivery system significantly accelerated wound healing by tuning the balance between the numbers of iNOS+CD206- M1 and iNOS-CD206+ M2 macrophages in the wound microenvironment.

## 2. Materials and Methods

### 2.1. Reagents

Oligochitosan (Och), inositol hexaphosphate (IP6), Alcian Blue (AB), Proanthocyanidins (PAs), Oil Red O (ORO), and Light Green SF (LG) were purchased from Sigma (St. Louis, MO, USA). LXA4 was purchased from Cayman Chemical (Michigan, USA). Except for Och and IP6, AB and PA were dissolved in ultrapure water, while ORO and LG were dissolved in alcohol solution according to the manufacturer’s instructions.

### 2.2. Animals

This study was approved by the Animal Care and Ethics Committee of Taipei Medical University. In this study, a full-thickness skin incision model was employed to evaluate the wound closure ability of ADC. All mouse experiments were conducted in accordance with the guidelines laid down by Taipei Medical University (LAC-2022-0161).

### 2.3. Preparation of Complex Coacervates

A quantity of 5 mg/mL Och and IP6 were dissolved in ultrapure water at various pH levels from 2.0 to 8.0 with 1M HCl or 1N NaOH solution, respectively. The stock solutions were sterile filtered (0.22 μm). The Och and IP6 solutions were mixed vigorously at various volume ratios of Och:IP6 ranging from 5:1 to 1:5. Liquid–liquid phase separation occurred immediately and the condensed liquid phase was settled down at room temperature within 30 min. The yields of ADC at various conditions were calculated by the following Equation (1):% yield = [(ADC0 − ADCi)/ADC0] × 100(1)
where ADC0 is the total powder weight of Och and IP6 used to make the ADC solutions and ADCi is the weight of the dried ADC after freeze-drying.

### 2.4. Turbidity Measurements

Turbidity at a wavelength of 580 nm was measured using spectrophotometry (Varioskan^®^ Flash, Thermo Scientific, Waltham, MA, USA) at room temperature. Neither Och nor IP6 absorb wavelengths at 580 nm. The Och and IP6 solutions were prepared at different pH conditions before being mixed. Before being mixed with IP6, different concentrations of NaCl were added to the Och solution. Once mixed with IP6, the solution was measured within 2 min.

### 2.5. Stability Measurements

ADC was prepared as previously described in ultrapure water. Specifically, the effect of pH on ADC stability was determined by preparing three batches of ADC for each time point and incubating them with 1 mL of buffered pH solution. Next, 1 mL supernatant from each group was replaced with 1 mL of fresh buffered pH solutions after 1, 2, 4, 8, 12, and 24 h, then every 24 h thereafter for 30 days. Since Och has a distinct UV absorbance at 274 nm, the Och concentration in the supernatants was measured at a wavelength of 274 nm using UV-vis spectrophotometry.

### 2.6. Encapsulation Procedure

Och was premixed vigorously with compounds AB, ORO, PA, and LG at various concentrations ranging from 20 mM to 1.25 mM. IP6 was then added to the premixed solution to form ADC. After 30 min of being settled down at room temperature, the supernatant was measured using UV-vis spectrometry since AB, ORO, PA, and LG have specific absorbances of 612 nm, 336 nm, 230 nm, and 611 nm, respectively, that do not overlap with Och at 274 nm. Determinations were performed in triplicate (see Appendix A).

The encapsulation efficiency of each compound was calculated using Equation (2):Encapsulation efficiency = (UV0/UVi) × 100(2)
where UV0 is the UV absorbance in the supernatant at a specific wavelength and UVi is the reference UV absorbance at known standard concentrations.

Regarding the encapsulation of LXA4 and Igs, we premixed Och with LXA4 and Igs separately, and then PA was added to the premixed solution. The solution was then vigorously mixed for 5 min before IP6 was added to form ADC. After 30 min of settling down, the supernatant was transferred to the ELISA microplates (Victor X4 2030, PerkinElmer, Waltham, MA, USA) according to the manufacturer’s instructions (PerkinElmer). UV absorbance at a 450 nm wavelength was measured from each sample and then normalized to its own standard curves. Samples were measured in triplicate from each condition.

### 2.7. Release Kinetics of PAs

An in vitro release study of PAs from ADC was performed with different buffered pH solutions. ADC-PA was prepared as previously described. Similar to the stability test of ADC, the effect of pH on the release kinetic of PAs was determined by UV-vis spectrometry at a wavelength of 230 nm. Briefly, supernatants were replaced with 1 mL of fresh buffered pH solutions after 1, 2, 4, 8, 12, and 24 h, then every 24 h thereafter for 30 days. The experiments were made in triplicate and average values were taken. The corresponding cumulative percent of PA that was released was determined using a standard calibration curve covering the range of the assay.

### 2.8. In Vitro Macrophages Culture and Stimulation

J774 cells were purchased from American Type Culture Collection (Manassas, VA, USA). Cells were grown under standard conditions in Dulbecco’s Modified Eagle Medium (DMEM) (Thermo Scientific, Massachusetts, USA), penicillin/streptomycin, and 10% fetal bovine serum (FBS) (Gibco) at 37 °C in a 5% CO_2_ atmosphere. To obtain bone marrow-derived macrophages (BMDMs), C57BL/6 mice were used. Bone marrow cells were flushed from the femurs and cultured for 7 days in 30% L929-cell-conditioned medium to obtain BMDMs. When J774 or BMDM cells reached 80% confluence, they were separately treated with phosphate-buffered saline (PBS), lipopolysaccharides (LPS) (100 ng/mL), Och, IP6, PA, ADC, and ADC-PA for 24 h and then fixed in 4% paraformaldehyde for later immunofluorescence analyses. To determine the inflammatory responses, both J774 and BMDM cells were separately treated with PBS, LPS, ADC, ADC+LPS, ADC-PA, and ADC+LPS for 6 h. Cells were then preserved in RNAzol solution (Molecular Research Center, Taipei, Taiwan) and stored at −80 °C before use.

### 2.9. Preparation of Thioglycollate-Elicited Peritoneal Macrophages

Brewer’s thioglycollate medium (Sigma) was prepared in distilled water at final 4% (*w*/*v*) and sterilized with an autoclave. A volume of 1 mL of the medium per mouse (total *n* = 5 mice) was injected into the intraperitoneal cavity. After 3 days, mice were sacrificed by carbon dioxide asphyxiation to collect peritoneal exudates cells using sterile PBS. Red blood cells were lysed using RBC lysis buffer (Thermo Fisher Scientific, Massachusetts, USA). Harvested cells were plated in 6-well plates. Cells were treated with the following: Vehicle (PBS), LPS, ADC+LPS, or ADC-PA+LPS for another 2 days. Cells were then collected for immunofluorescent analysis.

### 2.10. Immunofluorescence

Paraformaldehyde-fixed BMDM, peritoneal macrophages, and J774 from each group were washed with PBS and permeabilized with 0.1% Triton X-100/PBS at room temperature for 10 min. Samples were then blocked in 5% rabbit serum in PBS for 1 h at room temperature and incubated at 4 °C overnight with the following primary antibodies at 1:200 dilutions in 5% rabbit serum: rabbit anti-Ki67 (Abcam, Cambridge, UK), mouse-anti-iNOS (Abcam, Cambridge, UK), and rabbit anti-Arg-1 (Invitrogen, Waltham, MA, USA). Samples were washed three times for 5 min with PBS. They were then incubated for 1 h at room temperature with the following secondary antibodies at 1:400 dilutions in 5% rabbit serum: anti-rabbit Alexa Fluor 488 (Invitrogen), anti-mouse Alexa Fluor 594, and Alexa Fluor™ 594 Phalloidin at 1:40 dilutions (Invitrogen). Samples were then washed with PBS three times for 5 min each. Samples were then counter-stained with DAPI in Fluoroshield mounting medium (Sigma, St. Louis, MO, USA). The samples were analyzed by fluorescence microscopy (AXIO Imager D2, Zeiss) at a 20× magnification. A total depth of 20 µm was acquired for each sample and the total projection was visualized in the xy planes.

### 2.11. Quantitative Polymerase Chain Reaction (qPCR)

Total RNAs from BMDMs, peritoneal macrophages, and J774 were extracted using RNAzol (Molecular Research Center) according to the manufacturer’s instructions. Briefly, total RNAs were reverse transcribed into cDNAs with IQ2 MMLV RT-Script Synthesis Kit according to the manufacturer’s instructions (Bio-Genesis Technologies, South San Francisco, CA, USA). Total cDNAs were diluted at 1:50 ratios and used as templates for qPCR. The reactions were performed by adding the following reagents: 2.5 µL of each primer (stock 10 µM, see Appendix A), 5 µL of 1:50 cDNA dilutions, and 10 µL of 2 SYBR Green master mixes (Invitrogen). PCR conditions were performed on 96 well plates at the following temperature cycles: step 1: 95 °C for 5 min; step 2: 95 °C for 30 s, 60 °C for 30 s, and 72 °C for 35 s for 35 more cycles; step 3: 72 °C for 5 min. Melting curve detection was added at the final step to verify the specificity of the amplicons. Relative gene expressions were normalized using β-actin, and the results were plotted and analyzed using Prism 8 (GraphPad Software Inc., San Francisco, CA, USA).

### 2.12. Bioinformatic Analyses

PCA of the gene expression levels between each group in BMDM was quantified using qPCR. The statistical analysis used a logarithmic (log2) transformation of the data to stabilize the variance. The mean values of triplicate qPCR assays for each sample were analyzed statistically using the prcomp function in R (www.r-project.org, accessed on 20 October 2021). The PCA results are shown as the two-dimensional contribution scores for component numbers 1 and 2 (PC1 and PC2). The contribution scores were produced by conversion from each eigenvector value with 36 genes.

### 2.13. In Vivo Wound Healing in Diabetic Mice

A total of 18 male db/db mice were used as a model of type 2 diabetes. The mice with blood glucose levels higher than 250 mg/dL were confirmed as diabetic and used in this study. Two 6 mm circular full thickness excisional wounds were produced by medical punch on the dorsal side of each mouse according to the centerline along the back. Mice were randomly divided into three groups at day 0, 7, or 14 (*n* = 6 per group). Wounds on each mouse were either receiving the control (20 µL/mice, Nexcare liquid bandage, Minnesota, USA) or ADC-PA (20 µL/mice) for 14 consecutive days, with digital images of the wounds captured at days 0, 7, or 14 for all experimental mice. The wound areas were analyzed by tracing the wound margins and calculated using ImageJ 1.53u. The closure was expressed as a percentage of the original wound area. Percentage wound contraction was calculated as follows:% of Wound closure = (wound area on day zero − wound area on day n)/wound area on day zero × 100
where *n* = number of days (days 7 and 14).

### 2.14. Wound Immunofluorescence

Wounds were harvested from mice on days 7 or 14 following the creation of the wounds. Staining with macrophage markers iNOS and CD206 was used to evaluate the population of macrophages. Briefly, skin samples from the wounded areas and surrounding tissues (~1 cm in diameter, ~2 mm in thickness) were excised and immediately fixed in 10% formalin for 24 h at room temperature and subsequently embedded in paraffin. Following standard deparaffinization and antigen retrieval procedures, sections were blocked in 5% goat serum for 2 h at room temperature. The slides were then incubated with the primary antibodies anti-iNOS and anti-CD206 at 1:250 dilutions (Arigo Laboratories, USA) overnight at 4 °C. The next day, sections were washed and incubated with anti-goat or anti-mouse secondary antibodies (1:500 dilutions) at room temperature for 2 h. Sections were counter-stained with DAPI in Fluoroshield mounting medium (Sigma-Aldrich) and analyzed by fluorescence microscopy (AXIO Imager D2, Zeiss, Jena, Germany) at 20× magnification.

### 2.15. Collagen Quantification Using Masson’s Trichrome

The collagen content in diabetic wounds was stained with Masson’s Trichrome (Abcam) and the blue intensity was quantified using ImageJ software as previously described. Briefly, images were processed using color deconvolution to extract blue color, which was identified as collagen. The fixed threshold was then applied to each image and the intensity of blue color was analyzed.

### 2.16. Statistical Analyses

Data are presented as mean ± SD for the results from three or more experiments. *p*-values less than 0.05 were calculated from a two-tailed *t*-test or a two-way ANOVA with Prism (GraphPad Software Inc.) and taken to represent significant differences.

## 3. Results

### 3.1. Formation of a Complex Coacervation Base on Och and IP6 Is pH- and Salt-Dependent

Both Och and IP6 (Figure 1A) form complexes with other charged polyelectrolytes. Och is a weak base with a pKa around 6.2–7.0 [42,43,44,45,46], while the weak acid IP6 has a wide pKa range (3.2 to 12). Therefore, we would like to understand how the Och:IP6 mole ratio affects the generation of complex coacervates at different pH conditions. The phase diagram generated by varying the volume ratio between Och and IP6 from 5:1 to 1:5 at different pH conditions suggests that the highest yield (>85%) of complex coacervation is formed at pH 6.5 with a 2:1 ratio of Och/IP6 (Figure 1B). Decreased yields of ADC formation (<50%) were observed at pH 4.0, pH 6.0, and pH 7.0 (Figure 1A), with no ADC formation observed at pH 2.0, 7.5, and 8.0 (Figure 1B). Thus, the results suggest that the charge density of IP6 is important for ADC formation.

To visualize the morphology of ADC formation at various pH conditions (pH 2.0 to 8.0), fluorescent microscopes were used because of the autofluorescent property of ADC (Figure 1C). Consistent with the results of the phase diagram (Figure 1B), the green autofluorescent signal was observed when ADC formed at pH levels ranging from 4.0 to 7.0 with a 2:1 ratio of Och and IP6 (Figure 1C). Moreover, we found a homogeneous colloidal with micron size of ADC particles formed at pH 7.5 (Figure 1C). No fluorescent signal was observed at extreme pH levels, at pH 2.0 and 8.0 (Figure 1C), again suggesting that ADC formed at pH levels ranging from 4.0 to 7.0 with a 2:1 ratio of Och and IP6.

The formation of complex coacervates is not only associated with pH conditions but also affected by ionic strength [47]. In order to evaluate the effect of salt concentration on the formation of ADC, we varied the pH and salt concentrations by determining the turbidity at 580 nm, because neither Och nor IP6 absorb light at this wavelength. Our results showed that ionic strength has a strong effect on the formation of ADC, which is similar to most reported complex coacervate systems. Specifically, we found that the formation of ADC was most stable at salt concentrations below 400 mM at pH 6.0 and 6.5 (Figure 1D) and became unstable at salt concentrations above 400 mM (Figure 1D). Furthermore, when pH gradually increased (i.e., 7.0, 7.5, and 8.0), the ionic strength had a greater impact on the formation of ADC, making ADC significantly unstable at relatively low salt concentrations below 400 mM (Figure 1D).

### 3.2. Macroscopic Observation and Stability of ADC

Complex coacervates exhibit several interesting features, such as low interfacial tension and high loading capacity, that make them an excellent platform for delivering bioactives locally. To characterize whether ADC could be applied in physiological conditions (i.e., diabetic wounds), we first evaluated whether ADC could be applied under wet conditions. Our results showed that ADC is pipetable and can adhere to plastic and glass underwater (Figure 2A). Although the adhesion mechanism of Och requires further investigation, it is likely that both electrostatic (i.e., amine groups) and non-electrostatic (i.e., acetyl groups) adhesion mechanisms are involved due to the presence of 10% acetyl groups in Och. Furthermore, when ADC is applied underwater, it behaves as a dense liquid that can gradually be spread homogeneously on the plastic surface underwater (Figure 2B). Monitoring at higher magnification, ADC starts as microdroplets and can be rapidly fused into a film-like structure (Figure 2C). Together, flowable and film-forming behaviors make ADC suitable for covering the irregular shapes of wounds.

Och, but not IP6, has a distinct absorbance that can be detected by using UV spectrometry at 274 nm (Appendix A). Therefore, we are able to access ADC’s stability in fluid environments with various pH conditions by measuring the absorbance of Och through UV spectroscopy. Since the effect of pH is critical for structure and the drug release profiles of drug carriers, measuring the absorbance of Och also allows us to evaluate whether ADC can be used as a drug carrier for medical applications. We measured the release of Och at various pH conditions (2.0–9.0) from day 0 to day 30 at 37 °C in the presence of normal saline (0.9% NaCl). Our results revealed that ADC is more stable at neutral pH levels and is unstable at alkaline pH levels (Figure 2D). Specifically, ADC is stable in a pH range from 4.0–6.5, under which only around 30% of Och was dissociated from ADC by day 30. In contrast, ADC is unstable in alkaline conditions (pH 8.0), under which 97% of Och was dissociated from ADC by day 30 (Figure 2D).

### 3.3. Drug Encapsulation and In Vitro Drug Release of ADC

Grape seeds are rich sources of antioxidants’ proanthocyanidins (PAs), which are among the most common subgroups of flavonoids. Dietary consumption of PAs is considered safe and beneficial for human health. The benefits of PAs are mainly attributed to their antioxidant, antimicrobial, and anti-inflammatory properties by attenuating oxidative stress, scavenging free radicals, and inhibiting ROS-induced apoptotic pathways [31,34,35,48,49,50]. Recently, PAs have being used for the treatment of chronic wounds in vivo [51,52]. However, poor bioavailability (<10%) and instability (easily oxidized) significantly limit their application [53,54,55]. Thus, delivery efforts should be focused on improving the solubility of PAs and controlling their release. There is increasing evidence demonstrating the efficacy of coacervation as the basis of a drug delivery platform, taking advantage of the flexible and modular capabilities of charge-driven self-assembly to address each of these challenges [20,21,28,56]. Here, we demonstrated that ADC is capable of encapsulating PAs at a high yield (>86% encapsulation yield) when we used UV/VIS absorbance to evaluate encapsulation efficiency (Appendix A).

Moreover, complex coacervation techniques have been utilized for encapsulating hydrophobic compounds such as oils to enhance flavors in the food industry [23,57,58]. However, the capability of ADC to encapsulate a range of compounds is yet to be determined. Therefore, we tested the ability of ADC to encapsulate other small molecules with various chemical properties, such as Alcian Blue (AB, positively charged), Oil Red O (ORO, hydrophobic), and Light Green SF (LG, negatively charged). Our results show that ADC can rapidly encapsulate these compounds in one step with high efficiency (Figure 3A, Appendix A). These results further showed one of the major advantages of ADC formation based on complex coacervation: it has the ability to rapidly assemble in aqueous solutions that actively sequester protected drugs, regardless of the drug’s chemical nature.

Next, we further evaluated the drug release profiles of PAs from the ADC system. Our results show that ADC encapsulated with PAs (ADC-PA) exhibits different release profiles at various pH conditions (Figure 3B). The burst-release profile of ADC-PA was found in an acidic environment (pH 2.0), in which 50% of the PA was released in 7 days (Figure 3B). In contrast, ADC-PA showed a sustained release at neutral pH levels (4.0, 6.0, and 7.5), whereas 50% of the PAs were released in 17 days (Figure 3B). Furthermore, the release of ADC-PA was found to be most stable at alkaline pH levels (pH 8.0 and 9.0), at which less than 50% of PAs were released in 30 days (Figure 3B).

### 3.4. Cytotoxicity and Biocompatibility of ADC and ADC-PA

Recently, there has been increasing interest in designing novel biomaterial to modulate the activities of macrophages for chronic wound healing [59,60,61,62]. However, the molecular mechanism of macrophage behavior in response to complex coacervates (i.e., ADC or ADC-PA) remains largely unexplored. Previous studies have demonstrated that Och and IP6 are non-toxic to mammalian cells, including macrophages [36,39]. Moreover, it has been reported that PAs do not reduce macrophage viability after 24 h of exposure [63].

Little is known about the cytotoxic effect of Och, IP6, and PAs in the form of complex coacervates on nonactivated primary immune cells, such as primary bone marrow-derived macrophages (BMDMs). Here, we tested if Och and IP6, with or without encapsulated PAs, in the form of coacervates have cytotoxic effects on nonactivated BMDMs. In our previous study, we demonstrated that at a concentration of 200 µM, there is no toxic effect of IP6 [39]. Therefore, we used 10 μL each of ADC and ADC-PA (containing 200 μM IP6, 82 μM PA, and 500 μM CS) to investigate their effects on the proliferation of BMDMs. The concentration used in this study for ADC formation was non-toxic to immune cells that have been demonstrated (Figure 4 and Appendix A). Antibodies against the Ki67 protein have been widely used as proliferation markers for the detection of dividing cells. A strong, positive Ki67 staining in the cell nuclei suggested that the cells were dividing into new cells. Our results demonstrate that BMDMs treated with ADC or ADC-PA have a similar number of Ki67-positive cells as cells treated with PBS only, suggesting that both ADC and ADC-PA do not affect the proliferation of BMDMs (Figure 4A,B). We also examined whether ADC or ADC-PA had the same effect on the murine macrophage cell line J774A.1 using the same concentrations of ADC or ADC-PA that were used for BMDMs (primary cells). Similar results were observed in J774A.1 cells, showing that neither ADC nor ADC-PA affect the cell proliferation of J774A.1 cells (Appendix A).

### 3.5. The Effect of ADC and ADC-PA on the Modulation of LPS-Induced Pro-Inflammatory Responses and Resolution of Inflammation-Associated Gene Expression in Murine Macrophages

LPS has been extensively used to determine the anti-inflammatory and oxidative stress potential of test compounds. It is known to induce BMDM polarization into an M1 pro-inflammatory type by increasing several pro-inflammatory factors, such as tumor necrosis factor (TNFa), interleukin-6 (IL-6), interleukin-1 beta (IL-1b), and inducible nitric oxide synthase (iNOS) [64,65]. We therefore assessed the anti-inflammatory activity of Och and IP6, with or without encapsulated PAs, i.e., ADC or ADC-PA. BMDMs were stimulated with 100 ng/mL of LPS. Our results demonstrated that ADC alone reduced proinflammatory responses in BMDMs when exposed to 6 h of LPS stimulation (Figure 5A). The mRNA expression levels of *Tnfa*, *Il-1β*, *Il-6*, and *inos* were significantly further reduced when BMDMs were treated with ADC-PA in the presence of LPS, suggesting that the anti-inflammatory activity of BMDMs was further enhanced by ADC-PA compared to ADC (Figure 5A). Similar results were observed in J774A.1, in which ADC-PA further reduced *Il-1β* and *Il-6* mRNA levels in the presence of LPS compared to ADC-treated macrophages (Appendix A).

Macrophages are highly plastic cells that can polarize to either a pro-inflammatory M1 phenotype or an anti-inflammatory M2 phenotype in response to signals originating from the environment. To test whether ADC or ADC-PA affect macrophage phenotypic polarization by altering the gene expression profiles of macrophages in the presence of LPS, we examined the mRNA expression profiles of M2 macrophage marker genes. Although ADC can reduce the mRNA expression levels of pro-inflammatory cytokines such as *Tnfa*, *Il-1β*, *Il-6*, and *inos*, we did not observe any change in the mRNA levels of the M2 marker, with or without LPS. In contrast, *Tgfβ*, *Arg-1*, and *Egr-2* genes were significantly upregulated when BMDMs were treated with ADC-PA (Figure 5B). Even in the presence of an LPS stimulus, the ADC-PA-treated BMDMs had increased the mRNA levels of the M2 marker compared to that observed in the ADC-treated BMDMs, with or without LPS (Figure 5B). The M2-polarizing effects of ADC-PA remained effective after 48 h, indicating ADC is an effective drug delivery platform for controlled release (Figure 6).

The results of the genes expressed in BMDMs with different treatments were further analyzed by principal component analysis (PCA), and this demonstrated that ADC-PA-treated BMDMs were separated into a cluster that was transcriptionally distinct from other treatment groups (Figure 7). Our results reveal that ADC-PA exhibited preferential induction of genes associated with anti-inflammation and the resolution of inflammation pathways (Figure 7A), even in the presence of LPS, suggesting that ADC-PA can finely tune macrophage functions in an inflammatory environment triggered by LPS.

Similar to our BMDM results, the treatment of ADC or ADC-PA on peritoneal macrophages (Figure 8A) showed a significantly increased population of M2 macrophages (i.e., increased Arg-1 signal) and reduced M1 macrophages (i.e., decreased iNOS signal). The ratio of M2:M1 was significantly higher in the ADC-PA-treated group compared to the vehicle and LPS-stimulated groups (Figure 8B,C). Moreover, the effect of ADC-PA remains effective under the influence of LPS stimulation compared to the vehicle and LPS-stimulated groups, with ADC-PA-activated macrophages characterized by an anti-inflammatory profile (Figure 8C).

### 3.6. ADC-PA Facilitates Re-Epithelialization and Accelerates the Wound-Healing Process of Diabetic Mice

Mouse models of diabetes, such as db/db and NOD/ShiLtJ, which consistently have impaired wound healing, were used to investigate the effect of ADC-PA on the wound-healing process. On day 0, we generated full-thickness excisional wounds with a 7 mm biopsy punch (Figure 9A). The control and release data suggested that ADC-PA at neutral or higher pH levels exhibited sustained release profiles (Figure 3B), and therefore ADC-PA was administered to the wound site on days 0 and 7 compared to the untreated control (commercialized liquid bandage solution, Nexcare). Our results indicate that ADC-PA improved wound closure of 7 mm excisional diabetic mouse wounds compared to non-treated control wounds (Figure 9B and Appendix A). The rate of ADC-PA-treated wound closure was significantly faster at day 14, but not at day 7 (Figure 9C). We utilized Masson’s Trichrome to further evaluate the effect of ADC-PA on the wound-healing process. In contrast to tissue sections from control wounds, blue stains were more visible in the tissue sections from wounds treated with ADC-PA (Figure 9D and Appendix A), suggesting that deposition levels of collagen were more robust during the wound-healing process when wounds were treated with ADC-PA, facilitating epithelialization and wound healing (Figure 9E). Our results also showed that images of Masson’s Trichrome-stained tissue sections from wounds treated with ADC-PA had improved wound re-epithelialization. Adipose tissues and hair follicles could also be seen in tissue sections from wounds treated with ADC-PA (Figure 9D). Non-obese diabetic NOD/ShiLtJ mice showed similar results when their wounds were treated with ADC-PA (Appendix A). Together, the results suggest that ADC-PA shows beneficial effects for chronic wound healing.

### 3.7. Treatment with ADC-PA Leads to a Fine Balance between the Numbers of iNOS+CD206- M1 and iNOS-CD206+ M2 Macrophages in Wound Microenvironments

Chronic wounds such as diabetic wounds exhibit impaired healing that could be caused by persistent M1 macrophage polarization within the wound beds [14,66]. Our gene expression results showed that ADC-PA can finely tune macrophage functions in an inflammatory environment (Figure 5, Figure 6, Figure 7 and Figure 8 and Appendix A); therefore, we hypothesized that ADC-PA could resolve the impaired healing by reducing the population of M1 macrophages and increasing the M2 population. Since our results suggest that ADC-PA polarizes peritoneal macrophages toward an M2 subtype (Figure 5, Figure 6, Figure 7 and Figure 8), we next investigated whether treatment of ADC-PA affects the sub-populations of macrophages in chronic diabetic wound beds. Our results demonstrate that although the rate of ADC-PA-treated wound closure was significantly faster at day 14, but not at day 7, the ratios of iNOS+CD206- M1 macrophages:iNOS-CD206+ M2 macrophages were significantly different on day 7 and day 14 (Figure 10). Interestingly, ADC-PA-treated wounds at day 7 showed a balanced number of iNOS+CD206- M1 macrophages and iNOS-CD206+ M2 macrophages, whereas untreated controls at day 7 displayed high numbers of iNOS+CD206- M1 macrophages (Figure 10B). The ratio of M2:M1 macrophages under the effects of ADC-PA increased at day 14, when a lower number of iNOS+CD206- M1 macrophages was observed. In contrast, untreated controls at day 14 still displayed high numbers of iNOS+CD206- M1 macrophages (Figure 10B). Consistent with our in vitro results (Figure 5, Figure 6, Figure 7 and Figure 8), ADC-PA showed significant beneficial effects in accelerating diabetic wound healing via fine-tuning the balance between the numbers of M1 and M2 macrophages.

## 4. Discussion

Traditional delivery platforms, such as the delivery of small molecules using hydrogels, tend to suffer from the poor solubility of drugs and can be degraded and cleared rapidly from proteases and nucleases during delivery [67]. From a delivery standpoint, there is increasing evidence demonstrating the efficacy of complex coacervates as the basis of a drug delivery platform, taking advantage of the flexible and modular capabilities of charge-driven self-assembly to address each of these challenges [20,21].

Owing to its rapid and self-assembling properties, complex coacervation can encapsulate a wide range of drugs in a few simple steps without additional chemical modifications like most other drug carrier systems (i.e., hydrogels). Moreover, its flowable and film-forming abilities allow interconnecting into tissue interstices, increasing the surface area for maximum drug-release effects on wounded tissue [18,56]. In addition, drug encapsulation by complex coacervation needs neither any additive chemical crosslinkers nor any complex procedures. This makes complex coacervates an excellent platform for drug delivery. Such complexes have been widely applied in biomaterial [42,43,44,45]. However, there is a lack of studies investigating the use of Och and IP6 to generate a complex coacervation based on liquid–liquid phase separation and its applications. Unlike complexes formed by either Och or IP6 with other oppositely charged polyelectrolytes (i.e., dextran, gelatin, or glycosaminoglycans), the properties of complex coacervation based on liquid–liquid phase separation formed by Och and IP6 are unique. It is a highly condensed liquid phase of opposite-charged polyelectrolyte complex that displays low interfacial tension.

The formation of Och/IP6 complex coacervates is largely affected by pH and ionic strength, which is similar to other complex coacervates. As demonstrated in this study, the stability profile of ADC at pH 4.0–6.5 could be that the acidic-to-neutral pH conditions of Och do not affect the deprotonation of the amine groups of Och molecules (pKa ~6.5). In contrast, higher pH conditions caused the deprotonation of the amine groups, which in turn dissociated the integrity of the Och/IP6 network formed by the ionic interactions. Our results suggested that ADC provides long-term stability at a neutral pH and could be degraded by a pH-triggered mechanism, which is useful for medical applications. More importantly, the components of ADC are recognized as generally safe substances with low costs, and this affordability encourages scaling up for further medical applications.

Excessive ROS is associated with chronic inflammatory responses, which can greatly impair tissue regeneration [68]. Plant-based antioxidant PAs have been used to attenuate ROS, thus promoting wound-healing activities through facilitating re-epithelialization and tissue regeneration of damaged wounds [51,52]. Although popular in the nutraceuticals industry, its low bioavailability has hindered medical application and health benefits [53,54,55]. In this study, ADC was employed as a carrier to overcome the bioavailability limitations of PA. ADC-PA offers immune modulation in the wound-healing process, regulating the balance between M1 and M2 macrophages to accelerate diabetic wound healing (Figure 8 and Figure 10).

Non-healing wounds are associated with uncontrolled and sustained inflammatory responses (i.e., cytokines and ROS) in which pro-inflammatory M1 macrophages play a pivotal role in initiating these events by producing excessive ROS, as well as contributing to disease progression [12,65,66,69]. Therefore, targeting M1 macrophages and reprogramming them to the M2 healing type will be critical in the maintenance of immune system homeostasis and will provide a novel therapeutic approach for treating macrophage-associated inflammatory diseases. Although uncontrolled and sustained inflammation contributes to non-healing wounds, inflammation is also required in tissue and wound healing [12]. There is increasing evidence demonstrating that the ablation of macrophages during the early phases of repair or loss of TNFa hold back re-epithelialization and wound closure of murine cutaneous wounds [70,71]. Interestingly, one study has suggested that the early introduction of M2 macrophages during the early inflammatory phases disrupts the balance of the wound-healing process [72]. These results suggest that inflammation is essential in the wound-healing process, and the timing of the move from the inflammation to proliferative/maturation phases during the wound-healing process matters.

Our findings suggest that ADC-PA can facilitate the polarization of M2 macrophages, while not totally eradicating M1 macrophages in the wound bed (Figure 10). There is no significant difference between ADC-PA-treated mice and control mice in terms of the number of M1 macrophages on day 7. However, there is always a significant number of M2 macrophages in the wound beds of ADC-PA-treated mice compared to control mice (Figure 10B). The numbers of M1 and M2 macrophages are nearly equal in the early phase (day 7) of wound healing under the influence of ADC-PA (Figure 10B), indicating that some degree of initial pro-inflammatory response from M1 macrophages is necessary in order to activate the subsequent phases of healing. In contrast, the prolonged or excessive activation of inflammation from M1 macrophages without ADC-PA treatment is detrimental to wound healing (Figure 10).

Here, we showed that ADC is capable of polarizing M2 macrophages under inflammatory conditions, with the effects sustained owing to the controlled release provided by the ADC system in our in vitro and in vivo studies (Figure 11). Specifically, our in vitro drug release studies demonstrated that the release of ADC-PA was found to be most stable at alkaline pH levels, which is useful in treating chronic wounds due to the fact that acute wounds have a more neutral pH, while chronic wounds have an alkaline pH ranging from pH 7.15 to 8.9. Many previous studies have demonstrated that chronic wounds with such alkaline pH levels are associated with lower healing rates due to increased protease activity, resulting in a prolonged inflammatory state [59,60]. Together, our results suggest that ADC provides a sustained release profile in a physiological pH condition and also provides feasibility for designing a pH-sensitive drug release platform. Moreover, ADC is capable of encapsulating hydrophobic, hydrophilic, and IgG-based drugs at high yield (Figure 3, Appendix A), making it a unique type of drug carrier for antioxidants that can provide additional benefits to modulate the complex immune system.

## 5. Conclusions

In this study, we found that ADC-PA can induce macrophages to an M2 tissue-healing phenotype via up-regulation of anti-inflammatory and resolution of inflammatory responses. Our study in diabetic mouse wounds has shown that fine-tuning the balance between the numbers of M1 and M2 macrophages in the wound bed through the use of ADC-PA can significantly improve wound closure. This new type of ADC can potentially be extended beyond the current study and serve as a general drug delivery vehicle in other applications because it can be used as carrier in a diverse and multifaceted range of delivery systems, including tissue scaffold implants, capsules, films, or patches locally for a long period of time, depending on the medical purposes.

## Figures and Tables

**Figure 1 antioxidants-11-02351-f001:**
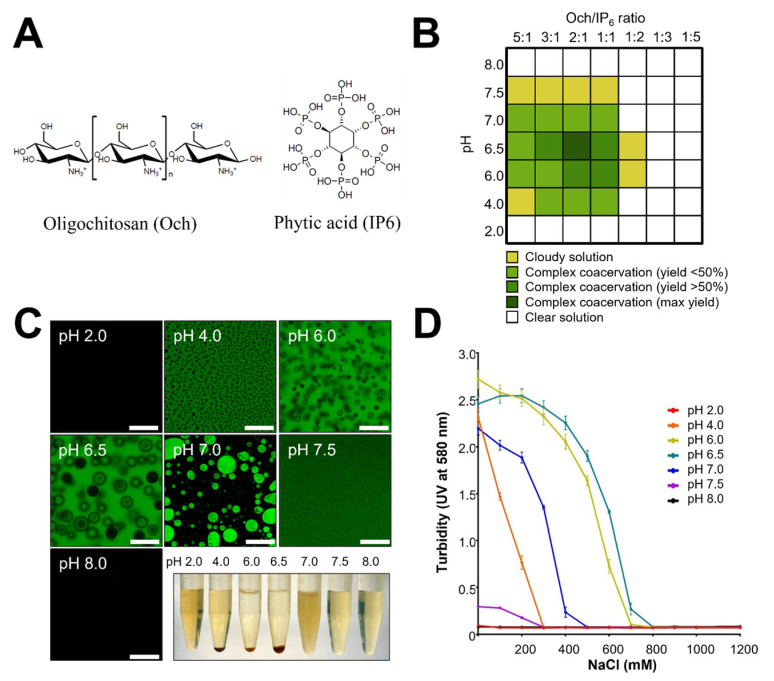
Formation of ADC based on complex coacervation. (**A**) Chemical structure of Och and IP6. (**B**) Effect of pH and molar ratio on the phase diagram of ADC complex formation as characterized by measuring the weight of formed ADC. (**C**) Fluorescence micrograph of ADC formation at different pH conditions. (**D**) ADC formation in the presence of salt and pH as characterized by UV spectrometer. Experiments were performed in triplicate. Scale bars = 20 µm.

**Figure 2 antioxidants-11-02351-f002:**
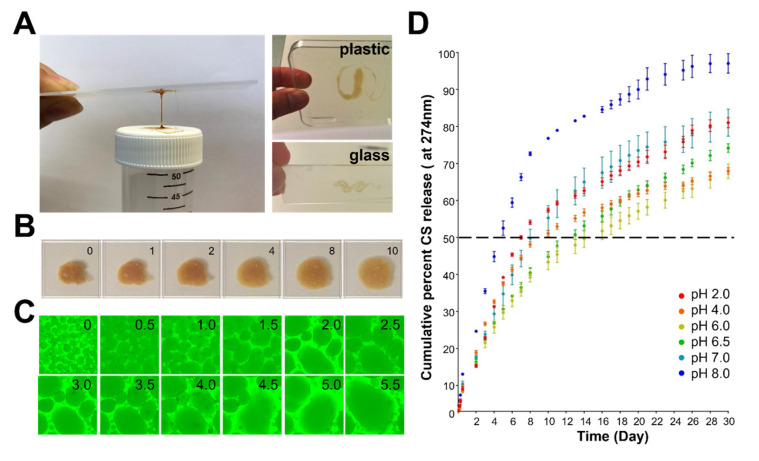
Properties of ADC. (**A**) ADC adhered to both glass and plastic surfaces. (**B**) Brightfield time-lapse photographs showed the liquid-like and flowable behavior of ADC underwater. Unit in minutes indicated by 0–10. (**C**) Time-lapse micrograph showed microdroplets fusing into a film-like structure of ADC in physiological conditions. Unit in minutes indicated by 0–5.5. (**D**) Stability of ADC at different pH conditions. Experiments were performed in triplicate.

**Figure 3 antioxidants-11-02351-f003:**
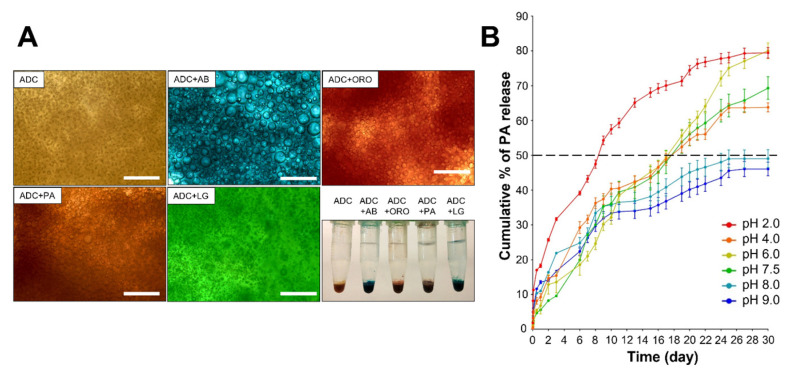
Drug encapsulation and release properties of ADC. (**A**) Different properties of compounds can be efficiently encapsulated into ADC. Abbreviations: ADC, adhesive drug carrier; AB, Alcian Blue; ORO, Oil Red O; PA, proanthocyanins; LG, Light Green SF. Scale bars = 50 µm. (**B**) PA release profiles at various pH conditions in the ADC system.

**Figure 4 antioxidants-11-02351-f004:**
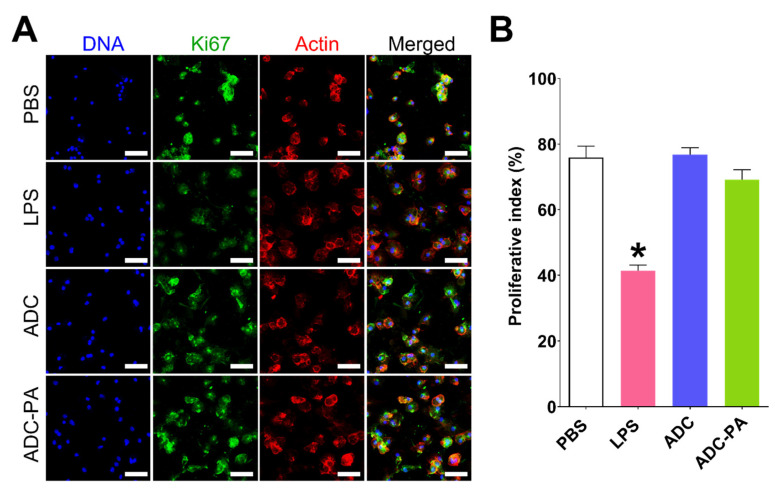
Treatment with ADC or ADC-PA does not affect the proliferation in nonactivated BMDM. (**A**) Immunofluorescence staining of the proliferation marker Ki67 in each group: Actin (red), Ki67 (green), and DNA (blue). The results from three independent experiments and representative images are shown. Scale bars = 50 μm. (**B**) Statistical analysis of Ki67-positive cells in each group. The results from six independent experiments are presented as the mean ± SD. * *p* < 0.05.

**Figure 5 antioxidants-11-02351-f005:**
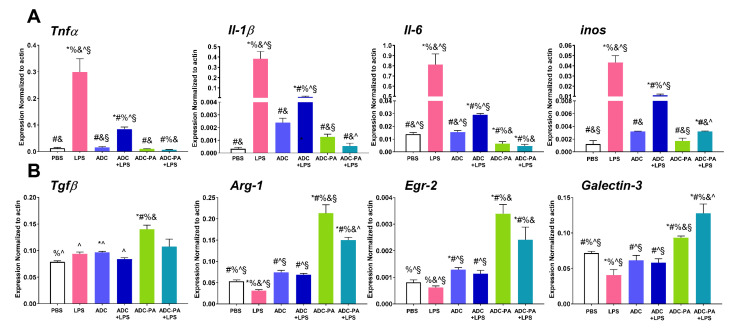
Treatment with ADC-PA alters LPS-mediated gene expression in BMDM. qPCR analysis of (**A**) pro-inflammatory cytokines and (**B**) anti-inflammatory cytokines in BMDMs treated with or without ADC or ADC-PA in the presence of LPS stimulation for 24 h. * indicates groups compared to PBS, # indicates groups compared to LPS, % indicates groups compared to ADC, & indicates groups compared to ADC+LPS, ^ indicated groups compared to ADC-PA, and § indicates groups compared to ADC-PA+LPS. The results from six independent experiments are presented as the mean ± SD. *, #, %, &, and § indicates *p* < 0.05.

**Figure 6 antioxidants-11-02351-f006:**
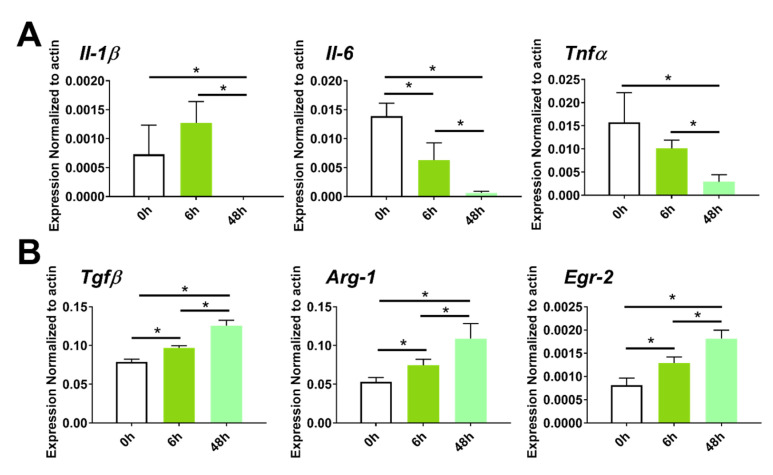
The long-term influence of ADC-PA on M2-associated gene expression of BMDM. qPCR analysis of (**A**) M1 and (**B**) M2-associated genes in BMDMs treated with ADC-PA at different time points (0 h, 6 h, and 48 h). The results from six independent experiments are presented as the mean ± SD. * *p* < 0.05, compared to 0 h.

**Figure 7 antioxidants-11-02351-f007:**
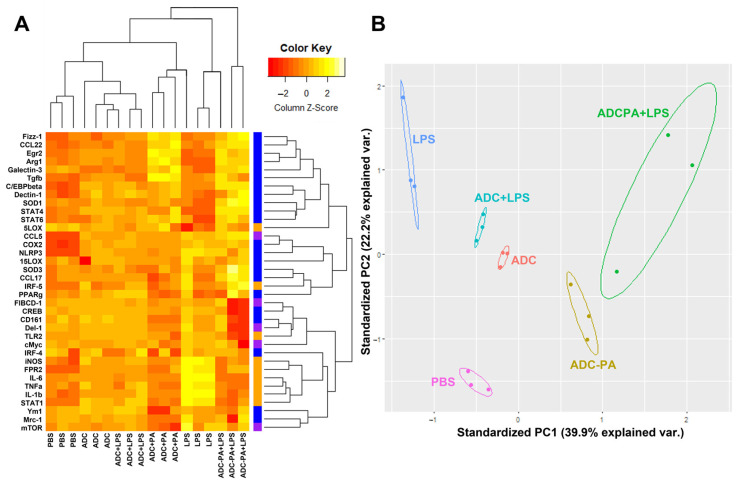
Induction of BMDM polarization by ADC-PA. (**A**) A heat map represents the relative expression levels of M1 and M2 markers in BMDM treated with different conditions: PBS, ADC, LPS, ACD+LPS, ADC-PA, and ADC-PA+LPS. The color scale represents the scaled abundance of each condition, with yellow indicating high abundance and red indicating low abundance. (**B**) Two-dimensional PCA (2D-PCA) score plots generated by analysis of each treatment condition, with circled regions illustrating the respective 95% confidence intervals. PC1 and PC2 describe 39.9% and 22.2% of the variance between the samples, respectively. Colored dots represent individual biological replicates. Experiments were conducted in triplicate.

**Figure 8 antioxidants-11-02351-f008:**
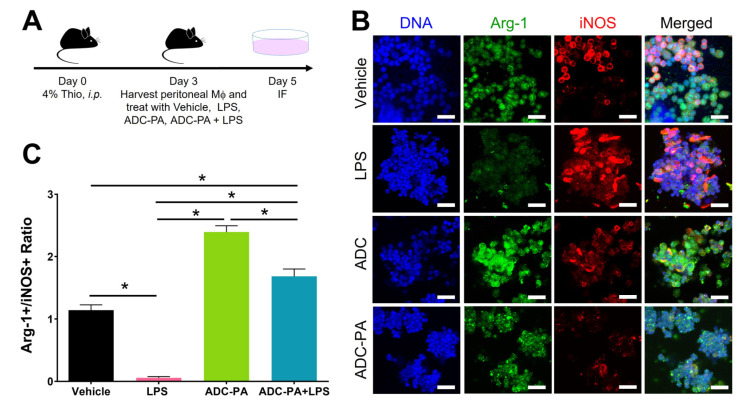
Ex vivo influence of ADC-PA on M2 macrophage polarization. (**A**) Schematic diagram showing the procedure to collect thioglycollate-elicited peritoneal macrophages from mice treated with different conditions. IF: immunofluorescence. (**B**) Immunofluorescence studies showed an increase of Arg-1-positive M2 macrophages and a decrease of iNOS-positive M1 macrophages in thioglycollate-elicited peritoneal macrophages from mice treated with ADC-PA. Scale bars = 50 μm. (**C**) Statistical graph showing relative expression ratio of Arg-1+/iNOS+ cells in different groups. Data are mean ± SEM of multiple fields in *n* = 5 per group. *, *p* < 0.05.

**Figure 9 antioxidants-11-02351-f009:**
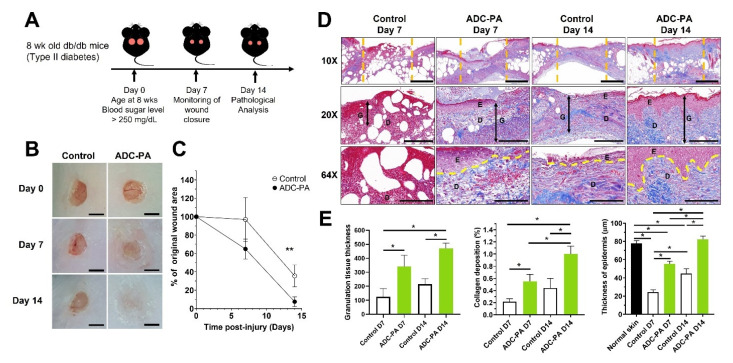
Healing effects of ADC-PA on full-thickness wounds in diabetic db/db mice. (**A**) Schematic diagram showing the procedure to generate a 7 mm full-thickness wound in db/db diabetic mice. (**B**) Representative images of wounds from each group over a 14-day period post-wounding. Nexcare liquid bandage solution was used as a positive control. Scale bar = 5 mm. (**C**) The graphical representation of the average wound closure in each group was measured using ImageJ software. (**D**) Masson’s Trichrome-stained skin tissue sections on days 7 and 14. Top panel scale bars = 1 mm. Middle panel scale bars = 200 μm. Lower panel scale bars = 100 μm. Orange dash lines indicate the wound site. Yellow dash lines distinguish the border between the epidermis and the dermis. Abbreviations: E, epidermis; D, dermis; G, granulation area. (**E**) Graphical representation of the expression of granulation thickness, collagen deposition, and epidermis thickness. The values are shown as mean ± SEM (*n* = 6 mice). * *p* < 0.05, ** *p* < 0.01, vs. the control group.

**Figure 10 antioxidants-11-02351-f010:**
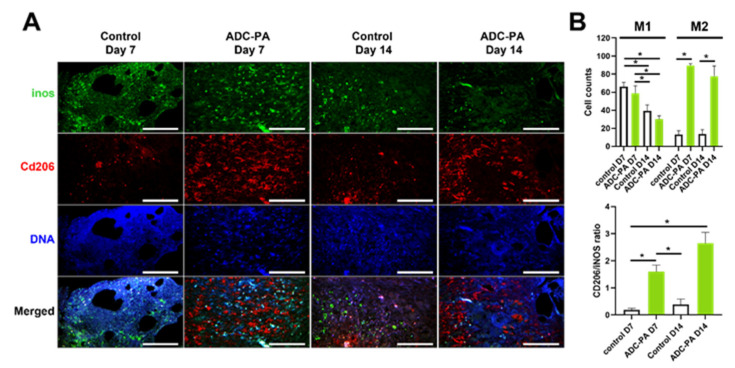
Significant macrophage polarization from M1 to M2 subtype occurs in the ADC-PA treated group. (**A**) Representative fluorescence micrographs showed wounded skin immunostained with M1 marker iNOS (green) and M2 marker CD206 (red) at days 7 and 14. Scale bars = 200 μm. (**B**) The number of M1 macrophages (iNOS^+^/CD206^−^) and M2 macrophages (Cd206^+^ iNOS^−^) per field in skin were statistically analyzed using ImageJ. The values are shown as mean ± SEM (*n* = 6 mice). * *p* < 0.05, vs. the control group. Abbreviations: iNOS, inducible nitric oxide synthase.

**Figure 11 antioxidants-11-02351-f011:**
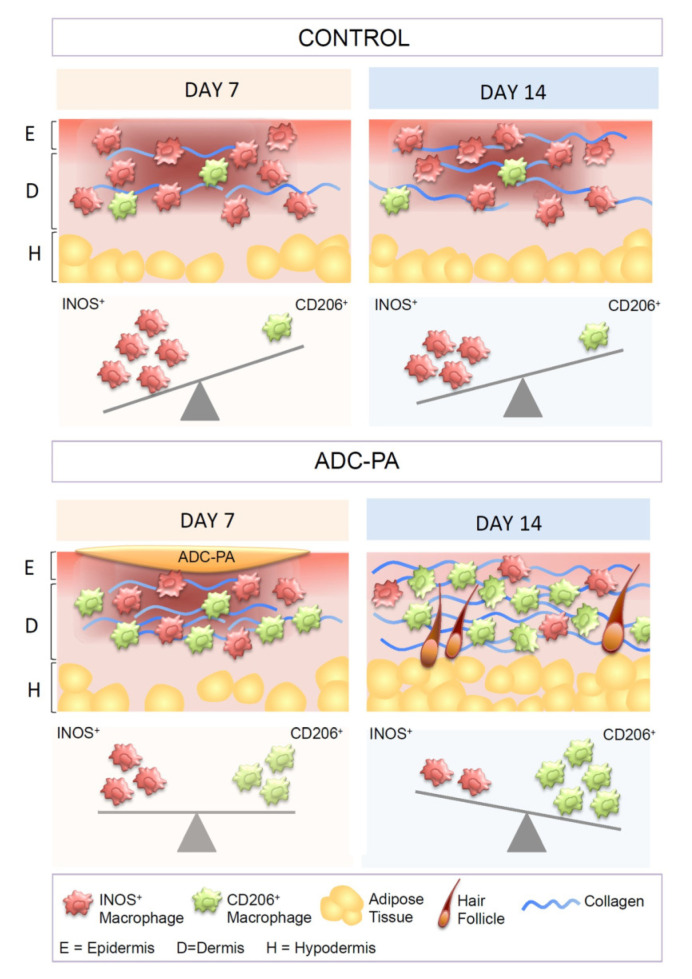
Schematic illustration of ADC-PA-guided macrophage reprogramming. ADC-PA treatment on wounds induces a conversion of M1 to M2 macrophages at an equilibrium phase and accelerates diabetic wound healing. Abbreviations: E, epidermis; D, dermis; H, hypodermis.

## Data Availability

The data presented in this study are available in this article and in the Appendix A.

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
