# Peer review of "Balance of Macrophage Activation by a Complex Coacervate-Based Adhesive Drug Carrier Facilitates Diabetic Wound Healing"

_antioxidants, 2022, doi:10.3390/antiox11122351_

Round 1

Reviewer 1 Report

Manuscript ID antioxidants-2020778

Balance of macrophage activation by a complex coacervate- 2 based adhesive drug carrier facilitates diabetic wound healing

Dear editor,

This is my report to you inform that the manuscript in this form is not suitable for publication.

In my opionion, this is an interesting manuscript in term of originality and Scientific Soundness.

The paper is generally well presented. However, in my opinion has some shortcomings in regards to some aspects, and I feel these  concepts have not been utilized to its full extent. More relevant and recent literature could be consulted.

Key critical points are

1.       Abstract is organized but extremely not very conclusive.

2.       Introduction section contains several information, while their relation to the current manuscript is not clearly illustrated.

3.       Materials and Methods should be expanded. The authors could better explain the methodology for Formation of a coacervation complex.

Given these shortcomings the manuscript requires major revisions.

Author Response

Dear Reviewers:

We really appreciate your thorough review of our proposed manuscript, and their valuable comments. We have carefully addressed the reviewers’ comments. Here, we have detailed the changes made to the manuscript and answered comments from the reviewers. We also rewrote the part with the most repetition as possible as we could. Please see the attached pdf file for more detail. Thank you again for your valuable time.

Reviewer 2 Report

Overall, I found this manuscript to be extremely interesting and well written. My main comment concerns the presentation of the results/discussion sections. When writing results, one must only include the results, with no no discussion. In sections 3.3, 3.4 and 3.5 the authors have melded results and discussion together. This is inappropriate. Please confine results to the results section only and any hypotheses, contextualisation of said results, please ensure that they are completely removed to the discussion section.

My general comments:

Abstract:

-lines 27 and 31: in vitro and in vivo are Latin and hence, should be italicised. Please amend

-line 34: iNOS, please write in full in the first instance and then abbreviate.

Introduction:

-line 50: hyphenate the words non and traumatic.

-line 57: delete the words " will be tempting" and replace with " is a high priority."

-line 61 at the end of the sentence (AEOS): add reference after "wound site" (ref).

-line 68 (AEOS): add reference after "chronic wounds" (ref).

-line 72 (first line/paragraph): add reference after "M2 macrophages."

-line 89: the word in situ is not hyphenated (please remove hyphen), but it is Latin, so please italicise.

-line 94 sentence beginning with (SBW): "Currently, there are only a handful of studies" Please add references to the end of the sentence (line 95).

-line 115: please italicise in vitro and in vivo

M&Ms:

-line 186: please italicise in vitro

-lines 196, 197 and 200: please define the following terms in the first instance and then abbreviate: DMEM, FBS, BMDMs, PBS and LPS. (NB: please ensure you define LPS when you first use the term as the abbreviation was written  earlier in the manuscript).

-line 209: Were all the mice given various treatments? What number of mice were administered the treatments? Please include information.

-line 210: How were the mice sacrificed? Please include information in this section.

-line 250: What total number of mice were used during this study? How many mice were allocated to each group? Please include information.

-line 263: The sentence "Wounds were harvested from mice on days 7 and 14 following creation of the wounds." This doesn't make any sense. Please rewrite.

Results:

-line 288 (SBW): "Such", through to the rest of the paragraph (line 298) is not a result! This whole section should be moved to the discussion. Please amend.

-line 332 (SBW): "Indeed" and rest of paragraph ending line 336 are again, not results. Please move section to discussion.

-line 368 (SBW): "The stability profile" through to line 374, not results, but discussion. As such, please move to discussion section.

Sections 3.3, 3.4 and 3.5: as stated in my overall comments above, the majority of these sections not results, but are, instead, discussing and contextualising the relevant results. Please move to discussion.

Discussion:

-1st line, 1st paragraph: the sentence is grammatically incorrect. Please amend.

-Why are you presenting new results in the discussion???! No, this is not appropriate. Please delete the new results (table 10 and associated text) in this section and transfer all to the results section where it belongs.

Author Response

(The authors gave the same response as above.)

Reviewer 3 Report

The authors prepared the manuscript titled Balance of macrophage activation by a complex coacervate- 2 based adhesive drug carrier facilitates diabetic wound healing. This paper is of importance since healing especially of chronic wounds is a challenge. There are a lot of results and a sufficient amount of methods. I have only several questions:

Did you test proanthocyanins in some experiments?

Why is necessary to measure turbidity in your experiments?

The abbreviation BMDMs should be introduced in the point 2.8., not on page 10.

There is a misspelling in line 259.

What do E, D, G mean in Figure 9D?

Did you test only ADC itself in mice wounds?

Author Response

(The authors gave the same response as above.)
